# Cold-Start Reinforcement Learning with Softmax Policy Gradient

**Nan Ding**
Google Inc.
Venice, CA 90291
dingnan@google.com

**Radu Soricut**
Google Inc.
Venice, CA 90291
rsoricut@google.com

## Abstract

Policy-gradient approaches to reinforcement learning have two common and undesirable overhead procedures, namely warm-start training and sample variance reduction. In this paper, we describe a reinforcement learning method based on a *softmax value function* that requires neither of these procedures. Our method combines the advantages of policy-gradient methods with the efficiency and simplicity of maximum-likelihood approaches. We apply this new cold-start reinforcement learning method in training sequence generation models for structured output prediction problems. Empirical evidence validates this method on automatic summarization and image captioning tasks.

## 1 Introduction

Reinforcement learning is the study of optimal sequential decision-making in an environment [16]. Its recent developments underpin a large variety of applications related to robotics [11, 5] and games [20]. Policy search in reinforcement learning refers to the search for optimal parameters for a given policy parameterization [5]. Policy search based on policy-gradient [26, 21] has been recently applied to structured output prediction for sequence generations. These methods alleviate two common problems that approaches based on training with the Maximum-likelihood Estimation (MLE) objective exhibit, namely the *exposure-bias* problem [24, 19] and the *wrong-objective* problem [19, 15] (more on this in Section 2). As a result of addressing these problems, policy-gradient methods achieve improved performance compared to MLE training in various tasks, including machine translation [19, 7], text summarization [19], and image captioning [19, 15].

Policy-gradient methods for sequence generation work as follows: first the model proposes a sequence, and the ground-truth target is used to compute a reward for the proposed sequence with respect to the reward of choice (using metrics known to correlate well with human-rated correctness, such as ROUGE [13] for summarization, BLEU [18] for machine translation, CIDEr [23] or SPICE [1] for image captioning, etc.). The reward is used as a weight for the log-likelihood of the proposed sequence, and learning is done by optimizing the weighted average of the log-likelihood of the proposed sequences. The policy-gradient approach works around the difficulty of differentiating the reward function (the majority of which are non-differentiable) by using it as a weight. However, since sequences proposed by the model are also used as the *target* of the model, they are very noisy and their initial quality is extremely poor. The difficulty of aligning the model output distribution with the reward distribution over the large search space of possible sequences makes training slow and inefficient[*]. As a result, overhead procedures such as warm-start training with the MLE objective and sophisticated methods for sample variance reduction are required to train with policy gradient.

---

[*]Search space size is $O(V^T)$, where $V$ is the number of word types in the vocabulary (typically between $10^4$ and $10^6$) and $T$ is the the sequence length (typically between 10 and 50), hence between $10^{40}$ and $10^{300}$.

The fundamental reason for the inefficiency of policy-gradient–based reinforcement learning is the large discrepancy between the model-output distribution and the reward distribution, especially in the early stages of training. If, instead of generating the target based solely on the model-output distribution, we generate it based on a proposal distribution that incorporates both the model-output distribution and the reward distribution, learning would be efficient, and neither warm-start training nor sample variance reduction would be needed. The outstanding problem is finding a value function that induces such a proposal distribution.

In this paper, we describe precisely such a value function, which in turn gives us a **S**oftmax **P**olicy **G**radient (SPG) method. The *softmax* terminology comes from the equation that defines this value function, see Section 3. The gradient of the softmax value function is equal to the average of the gradient of the log-likelihood of the targets whose proposal distribution combines both model output distribution and reward distribution. Although this distribution is infeasible to sample exactly, we show that one can draw samples approximately, based on an efficient forward-pass sampling scheme. To balance the importance between the model output distribution and the reward distribution, we use a bang-bang [8] mixture model to combine the two distributions. Such a scheme removes the need of fine-tuning the weights across different datasets and throughout the learning epochs. In addition to using a main metric as the task reward (ROUGE, CIDEr, etc.), we show that one can also incorporate additional, task-specific metrics to enforce various properties on the output sequences (Section 4). We numerically evaluate our method on two sequence generation benchmarks, a headline-generation task and an image-caption–generation task (Section 5). In both cases, the SPG method significantly improves the accuracy, compared to maximum-likelihood and other competing methods. Finally, it is worth noting that although the training and inference of the SPG method in the paper is mainly based on sequence learning, the idea can be extended to other reinforcement learning applications.

## 2  Limitations of Existing Sequence Learning Regimes

One of the standard approaches to sequence-learning training is Maximum-likelihood Estimation (MLE). Given a set of inputs $\mathbf{X} = \{\mathbf{x}^i\}$ and target sequences $\mathbf{Y} = \{\mathbf{y}^i\}$, the MLE loss function is:

$$L_{MLE}(\theta) = \sum_i L^i_{MLE}(\theta), \text{ where } L^i_{MLE}(\theta) = -\log p_\theta(\mathbf{y}^i | \mathbf{x}^i). \tag{1}$$

Here $\mathbf{x}^i$ and $\mathbf{y}^i = \{y^i_1, \ldots, y^i_T\}$ denote the input and the target sequence of the $i$-th example, respectively. For instance, in the image captioning task, $\mathbf{x}^i$ is the image of the $i$-th example, and $\mathbf{y}^i$ is the groundtruth caption of the $i$-th example.

Although widely used in many different applications, MLE estimation for sequence learning suffers from the exposure-bias problem [24, 19]. Exposure-bias refers to training procedures that produce brittle models that have only been exposed to their training data distribution but not to their own predictions. At training-time, $\log p_\theta(\mathbf{y}^i | \mathbf{x}^i) = \sum_t \log p_\theta(y^i_t | \mathbf{x}^i, \mathbf{y}^i_{1\ldots t-1})$, i.e. the loss of the $t$-th word is conditional on the true previous-target tokens $\mathbf{y}^i_{1\ldots t-1}$. However, since $\mathbf{y}^i_{1\ldots t-1}$ are unavailable during inference, replacing them with tokens $\mathbf{z}^i_{1\ldots t-1}$ generated by $p_\theta(\mathbf{z}^i_{1\ldots t-1} | \mathbf{x}^i)$ yields a significant discrepancy between how the model is used at training time versus inference time. The exposure-bias problem has recently received attention in neural-network settings with the "data as demonstrator" [24] and "scheduled sampling" [3] approaches. Although improving model performance in practice, such proposals have been shown to be statistically inconsistent [10], and still need to perform MLE-based warm-start training.

A more general approach to MLE is the Reward Augmented Maximum Likelihood (RAML) method [17]. RAML makes the correct observation that, under MLE, all alternative outputs are equally penalized through normalization, regardless of their relationship to the ground-truth target. Instead, RAML corrects for this shortcoming using an objective of the form:

$$L^i_{RAML}(\theta) = -\sum_{\mathbf{z}^i} r_R(\mathbf{z}^i | \mathbf{y}^i) \log p_\theta(\mathbf{z}^i | \mathbf{x}^i). \tag{2}$$

where $r_R(\mathbf{z}^i | \mathbf{y}^i) = \frac{\exp(R(\mathbf{z}^i | \mathbf{y}^i)/\tau)}{\sum_{\mathbf{z}^i} \exp(R(\mathbf{z}^i | \mathbf{y}^i)/\tau)}$. This formulation uses $R(\mathbf{z}^i | \mathbf{y}^i)$ to denote the value of a similarity metric $R$ between $\mathbf{z}^i$ and $\mathbf{y}^i$ (the reward), with $\mathbf{y}^i = \operatorname{argmax}_{\mathbf{z}^i} R(\mathbf{z}^i | \mathbf{y}^i)$; $\tau$ is a temperature hyper-parameter to control the peakiness of this reward distribution. Since the sum over all $\mathbf{z}^i$ for

the reward distribution $r_R(\mathbf{z}^i|\mathbf{y}^i)$ in Eq. (2) is infeasible to compute, a standard approach is to draw $J$ samples $\mathbf{z}^{i_j}$ from the reward distribution, and approximate the expectation by Monte Carlo integration:

$$L^i_{RAML}(\theta) \simeq -\frac{1}{J} \sum_{j=1}^{J} \log p_\theta(\mathbf{z}^{i_j}|\mathbf{x}^i). \tag{3}$$

Although a clear improvement over Eq. (1), the sampling for $\mathbf{z}^{i_j}$ in Eq. (3) is solely based on $r_R(\mathbf{z}^i|\mathbf{y}^i)$ and completely ignores the model probability. At the same time, this technique does not address the exposure bias problem at all.

A different approach, based on reinforcement learning methods, achieves sequence learning following a policy-gradient method [21]. Its appeal is that it not only solves the exposure-bias problem, but also directly alleviates the wrong-objective problem [19, 15] of MLE approaches. Wrong-objective refers to the critique that MLE-trained models tend to have suboptimal performance because such models are trained on a convenient objective (i.e., maximum likelihood) rather than a desirable objective (e.g., a metric known to correlate well with human-rated correctness). The policy-gradient method uses a value function $V_{PG}$, which is equivalent to a loss $L_{PG}$ defined as:

$$L^i_{PG}(\theta) = -V^i_{PG}(\theta), \ V^i_{PG}(\theta) = \mathbb{E}_{p_\theta(\mathbf{z}^i|\mathbf{x}^i)}[R(\mathbf{z}^i|\mathbf{y}^i)]. \tag{4}$$

The gradient for Eq. (4) is:

$$\frac{\partial}{\partial \theta} L^i_{PG}(\theta) = -\sum_{\mathbf{z}^i} p_\theta(\mathbf{z}^i|\mathbf{x}^i) R(\mathbf{z}^i|\mathbf{y}^i) \frac{\partial}{\partial \theta} \log p_\theta(\mathbf{z}^i|\mathbf{x}^i). \tag{5}$$

Similar to (3), one can draw $J$ samples $\mathbf{z}^{i_j}$ from $p_\theta(\mathbf{z}^i|\mathbf{x}^i)$ to approximate the expectation by Monte-Carlo integration:

$$\frac{\partial}{\partial \theta} L^i_{PG}(\theta) \simeq -\frac{1}{J} \sum_{j=1}^{J} R(\mathbf{z}^{i_j}|\mathbf{y}^i) \frac{\partial}{\partial \theta} \log p_\theta(\mathbf{z}^{i_j}|\mathbf{x}^i). \tag{6}$$

However, the large discrepancy between the model prediction distribution $p_\theta(\mathbf{z}^i|\mathbf{x}^i)$ and the reward $R(\mathbf{z}^i|\mathbf{y}^i)$'s values, which is especially acute during the early training stages, makes the Monte-Carlo integration extremely inefficient. As a result, this method also requires a warm-start phase in which the model distribution achieves some local maximum with respect to a reward-metric–free objective (e.g., MLE), followed by a model refinement phase in which reward-metric–based PG updates are used to refine the model [19, 7, 15]. Although this combination achieves better results in practice compared to pure likelihood-based approaches, it is unsatisfactory from a theoretical and modeling perspective, as well as inefficient from a speed-to-convergence perspective. Both these issues are addressed by the value function we describe next.

## 3 Softmax Policy Gradient (SPG) Method

In order to smoothly incorporate both the model distribution $p_\theta(\mathbf{z}^i|\mathbf{x}^i)$ and the reward metric $R(\mathbf{z}^i|\mathbf{y}^i)$, we replace the value function from Eq. 4 with a **S**oftmax value function for **P**olicy **G**radient (SPG), $V_{SPG}$, equivalent to a loss $L_{SPG}$ defined as:

$$L^i_{SPG}(\theta) = -V^i_{SPG}(\theta), \ V^i_{SPG}(\theta) = \log \left( \mathbb{E}_{p_\theta(\mathbf{z}^i|\mathbf{x}^i)}[\exp(R(\mathbf{z}^i|\mathbf{y}^i))] \right). \tag{7}$$

Because the value function for example $i$ is equal to $\text{Softmax}_{\mathbf{z}^i}(\log p_\theta(\mathbf{z}^i|\mathbf{x}^i) + R(\mathbf{z}^i|\mathbf{y}^i))$, where $\text{Softmax}_{\mathbf{z}^i}(\cdot) = \log \sum_{\mathbf{z}^i} \exp(\cdot)$, we call it the softmax value function. Note that the softmax value function from Eq. (7) is the dual of the entropy-regularized policy search (REPS) objective [5, 16] $L(q) = \mathbb{E}_q[R] + KL(q|p_\theta)$. However, our learning and sampling procedures are significantly different from REPS, as shown in what follows.

The gradient for Eq. (7) is:

$$\frac{\partial}{\partial \theta} L^i_{SPG}(\theta) = -\frac{1}{\sum_{\mathbf{z}^i} p_\theta(\mathbf{z}^i|\mathbf{x}^i)\exp(R(\mathbf{z}^i|\mathbf{y}^i))} \left( \sum_{\mathbf{z}^i} p_\theta(\mathbf{z}^i|\mathbf{x}^i)\exp(R(\mathbf{z}^i|\mathbf{y}^i)) \frac{\partial}{\partial \theta} \log p_\theta(\mathbf{z}^i|\mathbf{x}^i) \right)$$

$$= -\sum_{\mathbf{z}^i} q_\theta(\mathbf{z}^i|\mathbf{x}^i, \mathbf{y}^i) \frac{\partial}{\partial \theta} \log p_\theta(\mathbf{z}^i|\mathbf{x}^i) \tag{8}$$

where $q_\theta(\mathbf{z}^i|\mathbf{x}^i, \mathbf{y}^i) = \frac{1}{\sum_{\mathbf{z}^i} p_\theta(\mathbf{z}^i|\mathbf{x}^i)\exp(R(\mathbf{z}^i|\mathbf{y}^i))} p_\theta(\mathbf{z}^i|\mathbf{x}^i)\exp(R(\mathbf{z}^i|\mathbf{y}^i))$.

There are several advantages associated with the gradient from Eq. (8).

First, $q_\theta(\mathbf{z}^i|\mathbf{x}^i, \mathbf{y}^i)$ takes into account both $p_\theta(\mathbf{z}^i|\mathbf{x}^i)$ and $R(\mathbf{z}^i|\mathbf{y}^i)$. As a result, Monte Carlo integration over $q_\theta$-samples approximates Eq. (8) better, and has smaller variance compared to Eq. (5). This allows our model to start learning from scratch without the warm-start and variance-reduction crutches needed by previously-proposed PG approaches.

Second, as Figure 1 shows, the samples for the SPG method (pentagons) lie between the ground-truth target distribution (triangle and circles) and the model distribution (squares). These targets are both easier to learn by $p_\theta$ compared to ground-truth–only targets like the ones for MLE (triangle) and RAML (circles),

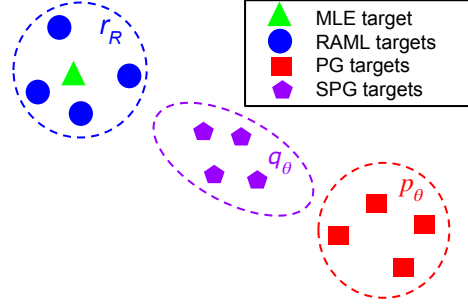

Figure 1: Comparing the target samples for MLE, RAML (the $r_R$ distribution), PG (the $p_\theta$ distribution), and SPG (the $q_\theta$ distribution).

and also carry more information about the ground-truth target compared to model-only samples (PG squares). This formulation allows us to directly address the exposure-bias problem, by allowing the model distribution to learn at training time how to deal with events conditioned on model-generated tokens, similar with what happens at inference time (more on this in Section 3.2). At the same time, the updates used for learning rely heavily on the influence of the reward metric $R(\mathbf{z}^i|\mathbf{y}^i)$, therefore directly addressing the wrong-objective problem. Together, these properties allow the model to achieve improved accuracy.

Third, although $q_\theta$ is infeasible for exact sampling, since both $p_\theta(\mathbf{z}^i|\mathbf{x}^i)$ and $\exp(R(\mathbf{z}^i|\mathbf{y}^i))$ are factorizable across $z_t^i$ (where $z_t^i$ denotes the $t$-th word of the $i$-th output sequence), we can apply *efficient* approximate inference for the SPG method as shown in the next section.

## 3.1 Inference

In order to estimate the gradient from Eq. (8) with Monte-Carlo integration, one needs to be able to draw samples from $q_\theta(\mathbf{z}^i|\mathbf{x}^i, \mathbf{y}^i)$. To tackle this problem, we first decompose $R(\mathbf{z}^i|\mathbf{y}^i)$ along the $t$-axis:

$$R(\mathbf{z}^i|\mathbf{y}^i) = \sum_{t=1}^{T} \underbrace{R(\mathbf{z}_{1:t}^i|\mathbf{y}^i) - R(\mathbf{z}_{1:t-1}^i|\mathbf{y}^i)}_{\triangleq \Delta r_t^i(z_t^i|\mathbf{y}^i, \mathbf{z}_{1:t-1}^i)},$$

where $R(\mathbf{z}_{1:t}^i|\mathbf{y}^i) - R(\mathbf{z}_{1:t-1}^i|\mathbf{y}^i)$ characterizes the *reward increment* for $z_t^i$. Using the reward increment notation, we can rewrite:

$$q_\theta(\mathbf{z}^i|\mathbf{x}^i, \mathbf{y}^i) = \frac{1}{Z_\theta(\mathbf{x}^i, \mathbf{y}^i)} \prod_{t=1}^{T} \exp(\log p_\theta(z_t^i|\mathbf{z}_{1:t-1}^i, \mathbf{x}^i) + \Delta r_t^i(z_t^i|\mathbf{y}^i, \mathbf{z}_{1:t-1}^i))$$

where $Z_\theta(\mathbf{x}^i, \mathbf{y}^i)$ is the partition function equal to the sum over all configurations of $\mathbf{z}^i$. Since the number of such configurations grows exponentially with respect to the sequence-length $T$, directly drawing from $q_\theta(\mathbf{z}^i|\mathbf{x}^i, \mathbf{y}^i)$ is infeasible. To make the inference efficient, we replace $q_\theta(\mathbf{z}^i|\mathbf{x}^i, \mathbf{y}^i)$ with the following approximate distribution:

$$\tilde{q}_\theta(\mathbf{z}^i|\mathbf{x}^i, \mathbf{y}^i) = \prod_{t=1}^{T} \tilde{q}_\theta(z_t^i|\mathbf{x}^i, \mathbf{y}^i, \mathbf{z}_{1:t-1}^i),$$

where

$$\tilde{q}_\theta(z_t^i|\mathbf{x}^i, \mathbf{y}^i, \mathbf{z}_{1:t-1}^i) = \frac{1}{\tilde{Z}_\theta(\mathbf{x}^i, \mathbf{y}^i, \mathbf{z}_{1:t-1}^i)} \exp(\log p_\theta(z_t^i|\mathbf{z}_{1:t-1}^i, \mathbf{x}^i) + \Delta r_t^i(z_t^i|\mathbf{y}^i, \mathbf{z}_{1:t-1}^i)).$$

By replacing $q_\theta$ in Eq. (8) with $\tilde{q}_\theta$, we obtain:

$$\frac{\partial}{\partial \theta} L_{SPG}^i(\theta) = -\sum_{\mathbf{z}^i} q_\theta(\mathbf{z}^i|\mathbf{x}^i, \mathbf{y}^i) \frac{\partial}{\partial \theta} \log p_\theta(\mathbf{z}^i|\mathbf{x}^i)$$

$$\simeq -\sum_{\mathbf{z}^i} \tilde{q}_\theta(\mathbf{z}^i|\mathbf{x}^i, \mathbf{y}^i) \frac{\partial}{\partial \theta} \log p_\theta(\mathbf{z}^i|\mathbf{x}^i) \triangleq \frac{\partial}{\partial \theta} \tilde{L}_{SPG}^i(\theta) \qquad (9)$$

Compared to $Z_\theta(\mathbf{x}^i, \mathbf{y}^i)$, $\tilde{Z}_\theta(\mathbf{x}^i, \mathbf{y}^i, \mathbf{z}^i_{1:t-1})$ sums over the configurations of one $z^i_t$ only. Therefore, the cost of drawing one $\mathbf{z}^i$ from $\tilde{q}_\theta(\mathbf{z}^i|\mathbf{x}^i, \mathbf{y}^i)$ grows only linearly with respect to $T$. Furthermore, for common reward metrics such as ROUGE and CIDEr, the computation of $\Delta r^i_t(z^i_t|\mathbf{y}^i, \mathbf{z}^i_{1:t-1})$ can be done in $O(T)$ instead of $O(V)$ (where $V$ is the size of the state space for $z^i_t$, i.e., vocabulary size). That is because the maximum number of unique words in $\mathbf{y}^i$ is $T$, and any words not in $\mathbf{y}^i$ have the same reward increment. When we limit ourselves to $J = 1$ sample for each example in Eq. (9), the approximate SPG inference time of each example is similar to the inference time for the gradient of the MLE objective. Combined with the empirical findings in Section 5 (Figure 3) where the steps for convergence are comparable, we conclude that the time for convergence for the SPG method is similar to the MLE based method.

## 3.2 Bang-bang Rewarded SPG Method

One additional difficulty for the SPG method is that the model's log-probability values $\log p_\theta(z^i_t|\mathbf{z}^i_{1:t-1}, \mathbf{x}^i)$ and the reward-increment values $R(\mathbf{z}^i_{1:t}|\mathbf{y}^i) - R(\mathbf{z}^i_{1:t-1}|\mathbf{y}^i)$ are not on the same scale. In order to balance the impact of these two factors, we need to weigh them appropriately. Formally, we achieve this by adding a weight $w^i_t$ to the reward increments: $\Delta r^i_t(z^i_t|\mathbf{y}^i, \mathbf{z}^i_{1:t-1}, w^i_t) \triangleq w^i_t \cdot \Delta r^i_t(z^i_t|\mathbf{y}^i, \mathbf{z}^i_{1:t-1})$ so that the total reward $R(\mathbf{z}^i|\mathbf{y}^i, \mathbf{w}^i) = \sum_{t=1}^T \Delta r^i_t(z^i_t|\mathbf{y}^i, \mathbf{z}^i_{1:t-1}, w^i_t)$. The approximate proposal distribution becomes $\tilde{q}_\theta(\mathbf{z}^i|\mathbf{x}^i, \mathbf{y}^i, \mathbf{w}^i) = \prod_{t=1}^T \tilde{q}_\theta(z^i_t|\mathbf{x}^i, \mathbf{y}^i, \mathbf{z}^i_{1:t-1}, w^i_t)$, where

$$\tilde{q}_\theta(z^i_t|\mathbf{x}^i, \mathbf{y}^i, \mathbf{z}^i_{1:t-1}, w^i_t) \propto \exp(\log p_\theta(z^i_t|\mathbf{z}^i_{1:t-1}, \mathbf{x}^i) + \Delta r^i_t(z^i_t|\mathbf{y}^i, \mathbf{z}^i_{1:t-1}, w^i_t)).$$

The challenge in this case is to choose an appropriate weight $w^i_t$, because $\log p_\theta(z^i_t|\mathbf{z}^i_{1:t-1}, \mathbf{x}^i)$ varies heavily for different $i, t$, as well as across different iterations and tasks.

In order to minimize the efforts for fine-tuning the reward weights, we propose a *bang-bang* rewarded softmax value function, equivalent to a loss $L_{BBSPG}$ defined as:

$$L^i_{BBSPG}(\theta) = -\sum_{\mathbf{w}^i} p(\mathbf{w}^i) \log \left( \mathbb{E}_{p_\theta(\mathbf{z}^i|\mathbf{x}^i)}[\exp(R(\mathbf{z}^i|\mathbf{y}^i, \mathbf{w}^i))] \right), \qquad (10)$$

$$\text{and} \quad \frac{\partial}{\partial\theta}\tilde{L}^i_{BBSPG}(\theta) = -\sum_{\mathbf{w}^i} p(\mathbf{w}^i) \underbrace{\sum_{\mathbf{z}^i} \tilde{q}_\theta(\mathbf{z}^i|\mathbf{x}^i, \mathbf{y}^i, \mathbf{w}^i)\frac{\partial}{\partial\theta}\log p_\theta(\mathbf{z}^i|\mathbf{x}^i)}_{\triangleq -\frac{\partial}{\partial\theta}\tilde{L}^i_{SPG}(\theta|\mathbf{w}^i)}, \qquad (11)$$

where $p(\mathbf{w}^i) = \prod_t p(w^i_t)$ and $p(w^i_t = 0) = p_{drop} = 1 - p(w^i_t = W)$. Here $W$ is a sufficiently large number (e.g., 10,000), $p_{drop}$ is a hyper-parameter in $[0, 1]$. The name *bang-bang* is borrowed from control theory [8], and refers to a system which switches abruptly between two extreme states (namely $W$ and 0).

When $w^i_t = W$, the term $\Delta r^i_t(z^i_t|\mathbf{y}^i, \mathbf{z}^i_{1:t-1}, w^i_t)$ overwhelms $\log p_\theta(z^i_t|\mathbf{z}^i_{1:t-1}, \mathbf{x}^i)$, so the sampling of $z^i_t$ is decided by the reward increment of $z^i_t$. It is important to emphasize that in general the groundtruth label $y^i_t \neq \text{argmax}_{z^i_t} \Delta r^i_t(z^i_t|\mathbf{y}^i, \mathbf{z}^i_{1:t-1})$, because $\mathbf{z}^i_{1:t-1}$ may not be the same as $\mathbf{y}^i_{1:t-1}$ (see an example in Figure 2). The only special case is when $p_{drop} = 0$, which forces $w^i_t$ to always equal $W$, and implies $z^i_t$ is always equal$^\dagger$ to $y^i_t$ (and therefore the SPG method reduces to the MLE method).

On the other hand, when $w^i_t = 0$, by definition $\Delta r^i_t(z^i_t|\mathbf{y}^i, \mathbf{z}^i_{1:t-1}, w^i_t) = 0$. In this case, the sampling of $z^i_t$ is based only on the model prediction distribution $p_\theta(z^i_t|\mathbf{z}^i_{1:t-1}, \mathbf{x}^i)$, the same situation we have at inference time. Furthermore, we have the following lemma (with the proof provided in the Supplementary Material):

| $t$ | 1 | 2 | 3 | 4 | 5 | 6 | 7 |
|---|---|---|---|---|---|---|---|
| $y_t$ | a | man | is | sitting | in | the | park |
| $w_t$ | W | W | W | 0 | W | ... | ... |
| $z_t$ | a | man | is | *in* | *the* | ... | ... |

$$\text{argmax } \Delta r_5(z_5|\mathbf{y}, \mathbf{z}_{1:4}) = \text{'the'} \neq y_5 = \text{'in'}$$

Figure 2: An example of sequence generation with the bang-bang reward weights. $z_4 =$ "$in$" is sampled from the model distribution since $w_4 = 0$. Although $w_5 = W$, $z_5 =$ "$the$" $\neq y_5$ because $z_4 =$ "$in$".

---

$^\dagger$This follows from recursively applying $R$'s property that $y^i_t = \text{argmax}_{z^i_t} \Delta r^i_t(z^i_t|\mathbf{y}^i, \mathbf{z}^i_{1:t-1} = \mathbf{y}^i_{1:t-1})$.

**Lemma 1** *When $w_t^i = 0$,*

$$\sum_{\mathbf{z}^i} \tilde{q}_\theta(\mathbf{z}^i|\mathbf{x}^i, \mathbf{y}^i, \mathbf{w}^i) \frac{\partial}{\partial\theta} \log p_\theta(z_t^i|\mathbf{x}^i, \mathbf{z}_{1:t-1}^i) = 0.$$

As a result, $\frac{\partial}{\partial\theta} \tilde{L}_{SPG}^i(\theta|\mathbf{w}^i)$ is very different from traditional PG-method gradients, in that only the $z_t^i$ with $w_t^i \neq 0$ are included. To see that, using the fact that $\log p_\theta(\mathbf{z}^i|\mathbf{x}^i) = \sum_{t=1}^T \log p_\theta(z_t^i|\mathbf{x}^i, \mathbf{z}_{1:t-1}^i)$,

$$\frac{\partial}{\partial\theta} \tilde{L}_{SPG}^i(\theta|\mathbf{w}^i) = -\sum_t \sum_{\mathbf{z}^i} \tilde{q}_\theta(\mathbf{z}^i|\mathbf{x}^i, \mathbf{y}^i, \mathbf{w}^i) \frac{\partial}{\partial\theta} \log p_\theta(z_t^i|\mathbf{x}^i, \mathbf{z}_{1:t-1}^i), \tag{12}$$

Using the result of Lemma 1, Eq. (12) is equal to:

$$\frac{\partial}{\partial\theta} \tilde{L}_{SPG}^i(\theta|\mathbf{w}^i) = -\sum_{\{t:w_t^i \neq 0\}} \sum_{\mathbf{z}^i} \tilde{q}_\theta(\mathbf{z}^i|\mathbf{x}^i, \mathbf{y}^i, \mathbf{w}^i) \frac{\partial}{\partial\theta} \log p_\theta(z_t^i|\mathbf{x}^i, \mathbf{z}_{1:t-1}^i)$$

$$= -\sum_{\mathbf{z}^i} \tilde{q}_\theta(\mathbf{z}^i|\mathbf{x}^i, \mathbf{y}^i, \mathbf{w}^i) \sum_{\{t:w_t^i \neq 0\}} \frac{\partial}{\partial\theta} \log p_\theta(z_t^i|\mathbf{x}^i, \mathbf{z}_{1:t-1}^i) \tag{13}$$

Using Monte-Carlo integration, we approximate Eq. (11) by first drawing $\mathbf{w}^{ij}$ from $p(\mathbf{w}^i)$ and then iteratively drawing $z_t^{ij}$ from $\tilde{q}_\theta(z_t^i|\mathbf{x}^i, \mathbf{z}_{1:t-1}^i, \mathbf{y}^i, w_t^{ij})$ for $t = 1, \ldots, T$. For larger values of $p_{drop}$, the $\mathbf{w}^{ij}$ sample contains more $w_t^{ij} = 0$ and the resulting $\mathbf{z}^{ij}$ contains proportionally more samples from the model prediction distribution (with a direct effect on alleviating the exposure-bias problem). After $\mathbf{z}^{ij}$ is obtained, only the log-likelihood of $z_t^{ij}$ when $w_t^{ij} \neq 0$ are included in the loss:

$$\frac{\partial}{\partial\theta} \tilde{L}_{BBSPG}^i(\theta) \simeq -\frac{1}{J} \sum_{j=1}^J \sum_{\{t:w_t^{ij} \neq 0\}} \frac{\partial}{\partial\theta} \log p_\theta(z_t^{ij}|\mathbf{x}^i, \mathbf{z}_{1:t-1}^{ij}). \tag{14}$$

The details about the gradient evaluation for the bang-bang rewarded softmax value function are described in Algorithm 1 of the Supplementary Material.

## 4   Additional Reward Functions

Besides the main reward function $R(\mathbf{z}^i|\mathbf{y}^i)$, additional reward functions can be used to enforce desirable properties for the output sequences. For instance, in summarization, we occasionally find that the decoded output sequence contains repeated words, e.g. "US R&B singer Marie Marie Marie Marie ...". In this framework, this can be directly fixed by using an additional auxiliary reward function that simply rewards negatively two consecutive tokens in the generated sequence:

$$\text{DUP}_t^i = \begin{cases} -1 & \text{if } z_t^i = z_{t-1}^i, \\ 0 & \text{otherwise.} \end{cases}$$

In conjunction with the bang-bang weight scheme, the introduction of such a reward function has the immediate effect of severely penalizing such "stuttering" in the model output; the decoded sequence after applying the DUP negative reward becomes: "US R&B singer Marie Christina has ...".

Additionally, we can use the same approach to correct for certain biases in the forward sampling approximation. For example, the following function negatively rewards the end-of-sentence symbol when the length of the output sequence is less than that of the ground-truth target sequence $|\mathbf{y}^i|$:

$$\text{EOS}_t^i = \begin{cases} -1 & \text{if } z_t^i = </\text{S}> \text{ and } t < |\mathbf{y}^i|, \\ 0 & \text{otherwise.} \end{cases}$$

A more detailed discussion about such reward functions is available in the Supplementary Material. During training, we linearly combine the main reward function with the auxiliary functions:

$$\Delta r_t^i(z_t^i|\mathbf{y}^i, \mathbf{z}_{1:t-1}^i, w_t^i) = w_t^i \cdot \left( R(\mathbf{z}_{1:t}^i|\mathbf{y}^i) - R(\mathbf{z}_{1:t-1}^i|\mathbf{y}^i) + \text{DUP}_t^i + \text{EOS}_t^i \right),$$

with $W = 10,000$. During testing, since the ground-truth target $\mathbf{y}^i$ is unavailable, this becomes:

$$\Delta r_t^i(z_t^i|\mathbf{y}^i, \mathbf{z}_{1:t-1}^i, W) = W \cdot \text{DUP}_t^i.$$

# 5 Experiments

We numerically evaluate the proposed softmax policy gradient (SPG) method on two sequence generation benchmarks: a document-summarization task for headline generation, and an automatic image-captioning task. We compare the results of the SPG method against the standard maximum likelihood estimation (MLE) method, as well as the reward augmented maximum likelihood (RAML) method [17]. Our experiments indicate that the SPG method outperforms significantly the other approaches on both the summarization and image-captioning tasks.

We implemented all the algorithms using TensorFlow 1.0 [6]. For the RAML method, we used $\tau = 0.85$ which was the best performer in [17]. For the SPG algorithm, all the results were obtained using a variant of ROUGE [13] as the main reward metric $R$, and $J = 1$ (sample one target for each example, see Eq. (14)). We report the impact of the $p_{drop}$ for values in $\{0.2, 0.4, 0.6, 0.8\}$.

In addition to using the main reward-metric for sampling targets, we also used it to weight the loss for target $\mathbf{z}^{i_j}$, as we found that it improved the performance of the SPG algorithm. We also applied a *naive* version of the policy gradient (PG) algorithm (without any variance reduction) by setting $p_{drop} = 0.0$, $W \to 0$, but failed to train any meaningful model with cold-start. When starting from a pre-trained MLE checkpoint, we found that it was unable to improve the original MLE result. This result confirms that variance-reduction is a requirement for the PG method to work, whereas our SPG method is free of such requirements.

## 5.1 Summarization Task: Headline Generation

Headline generation is a standard text generation task, taking as input a document and generating a concise summary/headline for it. In our experiments, the supervised data comes from the English Gigaword [9], and consists of news-articles paired with their headlines. We use a training set of about 6 million article-headline pairs, in addition to two randomly-extracted validation and evaluation sets of 10K examples each. In addition to the Gigaword evaluation set, we also report results on the standard DUC-2004 test set. The DUC-2004 consists of 500 news articles paired with four different human-generated groundtruth summaries, capped at 75 bytes.[‡] The expected output is a summary of roughly 14 words, created based on the input article.

We use the sequence-to-sequence recurrent neural network with attention model [2]. For encoding, we use a three-layer, 512-dimensional bidirectional RNN architecture, with a Gated Recurrent Unit (GRU) as the unit-cell [4]; for decoding, we use a similar three-layer, 512-dimensional GRU-based architecture. Both the encoder and decoder networks use a shared vocabulary and embedding matrix for encoding/decoding the word sequences, with a vocabulary consisting of 220K word types and a 512-dimensional embedding. We truncate

| Method | Gigaword-10K | DUC-2004 |
|---|---|---|
| MLE | $35.2 \pm 0.3$ | $22.6 \pm 0.6$ |
| RAML | $36.4 \pm 0.2$ | $23.1 \pm 0.6$ |
| SPG 0.2 | $36.6 \pm 0.2$ | $23.5 \pm 0.6$ |
| SPG 0.4 | $\mathbf{37.8 \pm 0.2}$ | $24.3 \pm 0.5$ |
| SPG 0.6 | $37.4 \pm 0.2$ | $24.1 \pm 0.5$ |
| SPG 0.8 | $37.3 \pm 0.2$ | $24.6 \pm 0.5$ |

Table 1: The F1 ROUGE-L scores (with standard errors) for headline generation.

the encoding sequences to a maximum of 30 tokens, and the decoding sequences to a maximum of 15 tokens. The model is optimized using ADAGRAD with a mini-batch size of 200, a learning rate of 0.01, and gradient clipping with norm equal to 4. We use 40 workers for computing the updates, and 10 parameter servers for model storing and (asynchronous and distributed) updating. We run the training procedure for 10M steps and pick the checkpoint with the best ROUGE-2 score on the Gigaword validation set.

We report ROUGE-L scores on the Gigaword evaluation set, as well as the DUC-2004 set, in Table 1. The scores are computed using the standard *pyrouge* package[§], with standard errors computed using bootstrap resampling [12]. As the numerical values indicate, the maximum performance is achieved when $p_{drop}$ is in mid-range, with 37.8 F1 ROUGE-L at $p_{drop} = 0.4$ on the large Gigaword evaluation set (a larger range for $p_{drop}$ between 0.4 and 0.8 gives comparable scores on the smaller DUC-2004 set). These numbers are significantly better compared to RAML (36.4 on Gigaword-10K), which in turn is significantly better compared to MLE (35.2).

---

[‡]This dataset is available by request at http://duc.nist.gov/data.html.

[§]Available at pypi.python.org/pypi/pyrouge/0.1.3

## 5.2   Automatic Image-Caption Generation

For the image-captioning task, we use the standard MSCOCO dataset [14]. The MSCOCO dataset contains 82K training images and 40K validation images, each with at least 5 groundtruth captions. The results are reported using the numerical values for the C40 testset reported by the MSCOCO online evaluation server[¶]. Following standard practice, we combine the training and validation datasets for training our model, and hold out a subset of 4K images as our validation set.

Our model architecture is simple, following the approach taken by the Show-and-Tell approach [25]. We use a one 512-dimensional RNN architecture with an LSTM unit-cell, with a dropout rate equal of 0.3 applied to both input and output of the LSTM layer. We use the same vocabulary size of 8,854 word-types as in [25], with 512-dimensional word-embeddings. We truncate the decoding sequences to a maximum of 15 tokens. The input image is embedded by first passing it through a pretrained Inception-V3 network [22], and then projected to a 512-dimensional vector. The model is optimized using ADAGRAD with a mini-batch size of 25, a learning rate of 0.01, and gradient clipping with norm equal to 4. We run the training procedure for 4M steps and pick the checkpoint of the best CIDEr score [23] on our held-out 4K validation set.

|        | Validation-4K |           | C40   |
|--------|---------------|-----------|-------|
| Method | CIDEr         | ROUGE-L   | CIDEr |
| MLE    | 0.968         | 37.7 ± 0.1 | 0.94 |
| RAML   | 0.997         | 38.0 ± 0.1 | 0.97 |
| SPG 0.2 | 1.001        | 38.0 ± 0.1 | 0.98 |
| SPG 0.4 | 1.013        | 38.1 ± 0.1 | 1.00 |
| SPG 0.6 | **1.033**    | **38.2 ± 0.1** | **1.01** |
| SPG 0.8 | 1.009        | 37.7 ± 0.1 | 1.00 |

Table 2: The CIDEr (with the *coco-caption* package) and ROUGE-L (with the *pyrouge* package) scores for image captioning on MSCOCO.

We report both CIDEr and ROUGE-L scores on our 4K Validation set, as well as CIDEr scores on the official C40 testset as reported by the MSCOCO online evaluation server, in Table 2. The CIDEr scores are reported using the *coco-caption* evaluation toolkit[‖], while ROUGE-L scores are reported using the standard *pyrouge* package (note that these ROUGE-L scores are generally lower than those reported by the coco-caption toolkit, as it reports an average score over multiple reference, while the latter reports the maximum).

The evaluation results indicate that the SPG method is superior to both the MLE and RAML methods. The maximum score is obtained with $p_{drop} = 0.6$, with a CIDEr score of 1.01 on the C40 testset. In contrast, on the same testset, the RAML method has a CIDEr score of 0.97, and the MLE method a score of 0.94. In

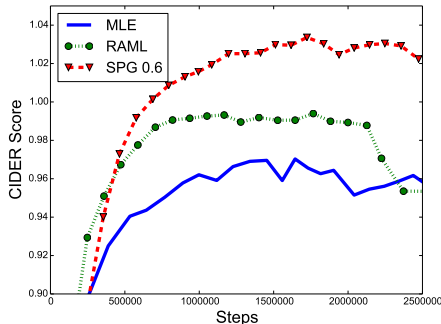

Figure 3: Number of training steps vs. CIDEr scores (on Validation-4K) for various learning regimes.

Figure 3, we show that the number of steps for SPG to converge is similar to the one for MLE/RAML. With the per-step inference cost of those methods being similar (see Section 3.1), the overall convergence time for the SPG method is similar to the MLE and RAML methods.

## 6   Conclusion

The reinforcement learning method presented in this paper, based on a softmax value function, is an efficient policy-gradient approach that eliminates the need for warm-start training and sample variance reduction during policy updates. We show that this approach allows us to tackle sequence generation tasks by training models that avoid two long-standing issues: the exposure-bias problem and the wrong-objective problem. Experimental results confirm that the proposed method achieves superior performance on two different structured output prediction problems, one for text-to-text (automatic summarization) and one for image-to-text (automatic image captioning). We plan to explore and exploit the properties of this method for other reinforcement learning problems as well as the impact of various, more-advanced reward functions on the performance of the learned models.

---

[¶]Available at http://mscoco.org/dataset/#captions-eval.

[‖]Available at https://github.com/tylin/coco-caption.

**Acknowledgments**

We greatly appreciate Sebastian Goodman for his contributions to the experiment code. We would also like to acknowledge Ning Ye and Zhenhai Zhu for their help with the image captioning model calibration as well as the anonymous reviewers for their valuable comments.

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
