[Supplementary Material]

# Supplementary Material: Cold-Start Reinforcement Learning with Softmax Policy Gradient

**Nan Ding**
Google Inc.
Venice, CA 90291
dingnan@google.com

**Radu Soricut**
Google Inc.
Venice, CA 90291
rsoricut@google.com

## 1 Bang-bang Rewarded SPG: Lemma 1

We provide here the proof for Lemma 1, as part of the derivation for the gradient computation method for the Bang-bang rewarded SPG method.

**Lemma 1** When $w_t^i = 0$,

$$\sum_{\mathbf{z}^i} \tilde{q}_\theta(\mathbf{z}^i|\mathbf{x}^i, \mathbf{y}^i, \mathbf{w}^i) \frac{\partial}{\partial \theta} \log p_\theta(z_t^i|\mathbf{x}^i, \mathbf{z}_{1:t-1}^i) = 0.$$

**Proof** First of all,

$$\tilde{q}_\theta(z_t^i|\mathbf{x}^i, \mathbf{y}^i, \mathbf{z}_{1:t-1}^i, w_t^i) \propto \exp\left(\log p_\theta(z_t^i|\mathbf{z}_{1:t-1}^i, \mathbf{x}^i) + \Delta r_t^i\right), \tag{1}$$

where $\Delta r_t^i = w_t^i \cdot (R(\mathbf{z}_{1:t}^i|\mathbf{y}^i) - R(\mathbf{z}_{1:t-1}^i|\mathbf{y}^i))$. When $w_t^i = 0$, $\Delta r_t^i = 0$, therefore,

$$\tilde{q}_\theta(z_t^i|\mathbf{x}^i, \mathbf{y}^i, \mathbf{z}_{1:t-1}^i, w_t^i) \propto \exp\left(\log p_\theta(z_t^i|\mathbf{z}_{1:t-1}^i, \mathbf{x}^i)\right) = p_\theta(z_t^i|\mathbf{z}_{1:t-1}^i, \mathbf{x}^i).$$

Therefore, the gradient component at time $t$ of example $i$ is:

$$\sum_{\mathbf{z}^i} \tilde{q}_\theta(\mathbf{z}^i|\mathbf{x}^i, \mathbf{y}^i, \mathbf{w}^i) \frac{\partial}{\partial \theta} \log p_\theta(z_t^i|\mathbf{x}^i, \mathbf{z}_{1:t-1}^i)$$

$$= \sum_{\mathbf{z}_{1:t}^i} \tilde{q}_\theta(\mathbf{z}_{1:t}^i|\mathbf{x}^i, \mathbf{y}^i, \mathbf{w}^i) \frac{\partial}{\partial \theta} \log p_\theta(z_t^i|\mathbf{x}^i, \mathbf{z}_{1:t-1}^i)$$

$$= \sum_{\mathbf{z}_{1:t-1}^i} \tilde{q}_\theta(\mathbf{z}_{1:t-1}^i|\mathbf{x}^i, \mathbf{y}^i, \mathbf{w}^i) \sum_{z_t^i} \tilde{q}_\theta(z_t^i|\mathbf{x}^i, \mathbf{y}^i, \mathbf{z}_{1:t-1}^i, w_t^i) \frac{\partial}{\partial \theta} \log p_\theta(z_t^i|\mathbf{x}^i, \mathbf{z}_{1:t-1}^i)$$

$$= \sum_{\mathbf{z}_{1:t-1}^i} \tilde{q}_\theta(\mathbf{z}_{1:t-1}^i|\mathbf{x}^i, \mathbf{y}^i, \mathbf{w}^i) \sum_{z_t^i} p_\theta(z_t^i|\mathbf{x}^i, \mathbf{z}_{1:t-1}^i) \frac{\partial}{\partial \theta} \log p_\theta(z_t^i|\mathbf{x}^i, \mathbf{z}_{1:t-1}^i)$$

$$= \sum_{\mathbf{z}_{1:t-1}^i} \tilde{q}_\theta(\mathbf{z}_{1:t-1}^i|\mathbf{x}^i, \mathbf{y}^i, \mathbf{w}^i) \sum_{z_t^i} \frac{\partial}{\partial \theta} p_\theta(z_t^i|\mathbf{x}^i, \mathbf{z}_{1:t-1}^i)$$

$$= \sum_{\mathbf{z}_{1:t-1}^i} \tilde{q}_\theta(\mathbf{z}_{1:t-1}^i|\mathbf{x}^i, \mathbf{y}^i, \mathbf{w}^i) \frac{\partial}{\partial \theta} \underbrace{\sum_{z_t^i} p_\theta(z_t^i|\mathbf{x}^i, \mathbf{z}_{1:t-1}^i)}_{= \frac{\partial}{\partial \theta} 1 = 0} = 0.$$

∎

## 2 Algorithm 1: Gradient for the Bang-bang Rewarded Softmax Value Function

The gradient computation for the Bang-bang Rewarded Softmax Value Function is formulated in Algorithm 1.

---

**Algorithm 1:** GRADIENT FOR THE BANG-BANG REWARDED SOFTMAX VALUE FUNCTION

---

**Input**: Data point $(\mathbf{x}^i, \mathbf{y}^i)$, hyperparameter $p_{drop}$, $W$, $J$, model parameter $\theta$.

**Result**: Gradient of data point $(\mathbf{x}^i, \mathbf{y}^i)$: $\frac{\partial}{\partial \theta} \tilde{L}^i_{BBSPG}(\theta)$.

$\frac{\partial}{\partial \theta} \tilde{L}^i_{BBSPG}(\theta) = 0$

**for** $j \in 1, \ldots, J$ **do**

    $\mathbf{z}^{i_j} \leftarrow \emptyset$

    **for** $t \in 1, \ldots, T$ **do**

        Sample $\mu_t^{i_j} \sim U[0, 1]$

        **if** $\mu_t^{i_j} > p_{drop}$ **then**

            $\Delta r_t^{i_j} = W \left( R(\mathbf{z}_{1:t}^{i_j}|\mathbf{y}^i) - R(\mathbf{z}_{1:t-1}^{i_j}|\mathbf{y}^i) + \text{DUP}_t^{i_j} + \text{EOS}_t^{i_j} \right)$

            Sample $z_t^{i_j} \sim \exp \left( \log p_\theta(z_t^{i_j}|\mathbf{z}_{1:t-1}^{i_j}, \mathbf{x}^i) + \Delta r_t^{i_j} \right) / Z$

            $\frac{\partial}{\partial \theta} \tilde{L}^i_{BBSPG}(\theta) = \frac{\partial}{\partial \theta} \tilde{L}^i_{BBSPG}(\theta) - \frac{\partial}{\partial \theta} \log p_\theta(z_t^{i_j}|\mathbf{x}^i, \mathbf{z}_{1:t-1}^{i_j})$

        **else**

            Sample $z_t^{i_j} \sim p_\theta(z_t^{i_j}|\mathbf{z}_{1:t-1}^{i_j}, \mathbf{x}^i)$

        **end**

        $\mathbf{z}^{i_j} \leftarrow \mathbf{z}^{i_j} \cup \left\{ z_t^{i_j} \right\}$.

    **end**

**end**

---

The reward functions used by the algorithm above are the ones discussed in Section 4 of the main paper. We extend that discussion in the section below.

## 3 Reward Functions for the SPG Method

### 3.1 Main Reward Function

In our experiments, the main reward metric is an average over ROUGE-1, ROUGE-2, and ROUGE-3 F1 scores. We choose ROUGE-$n$ [2] based on its good performance as an evaluation metric for both summarization and image-captioning, as well as because it is more computationally efficient compared to other scores such as CIDEr [4] or SPICE [1].

The reason we average up to $n = 3$ (instead of just 2) is illustrated in the following target example:

$$\text{a man is standing on a street } </S> \tag{2}$$

In the above sentence, the word 'a' appears twice. When using a ROUGE average up to $n = 2$ as the reward metric, for $z_{t-1} = $ 'a', both words 'man' and 'street' have identical reward increments. Therefore, this reward metric cannot distinguish between them. More generally, if the metric used does not account for n-grams longer than 2, it is suboptimal for decisions following common words (like 'the', 'of', or 'a').

### 3.2 ROUGE-L as a Reward Function

The ROUGE-L metric [2] also cannot be applied as the main reward metric by itself. Using Example (2) above, when $\mathbf{z}_{1:t-1} = $ 'a', all the remaining target words have identical reward increments under ROUGE-L, because the length of the longest-common-subsequences is the same for all (i.e., 2). Furthermore, if $\mathbf{z}_{1:t-1} = $ 'a street', all words (inside or outside the target) except '</S>' have a 0 reward increment value because it would not improve the length of the longest-common-subsequence.

Although not attempted in this paper, one may combine the ROUGE-L metric with other metrics, such as the one in Section 3.1 above. A similar proposal, albeit in a more traditional PG setting, has been made in [3], taking advantage of the additional signal provided by various metrics.

## 3.3 EOS Reward Function

In the main paper, we introduce an EOS reward function which negatively rewards the end-of-sentence symbol when the length of the output sequence is less than the length of the ground-truth target sequence $|\mathbf{y}^i|$:

$$\text{EOS}_t^i = \begin{cases} -1 & \text{if } z_t^i = \text{</S>} \text{ and } t < |\mathbf{y}^i|, \\ 0 & \text{otherwise.} \end{cases}$$

We illustrate the reason for this reward function using Example (2) again. If $\mathbf{z}_{1:t-1} = $ 'a street', then the word with the most reward increment is '</S>'. However, target sequence $\mathbf{z} = $ 'a street </S>' is too short and misses a lot information, since there are five remaining words in the ground-truth target that have not been exploited. The EOS function encourages the generation of longer sequences, by correcting the bias introduced by the greediness of the forward-pass sampling step.

## 3.4 Before/After Examples when using the DUP Reward Function

The DUP function penalizes consecutive tokens in the generated sequence, which helps alleviating "stuttering" in the model output. The use of the DUP function helps improving the ROUGE-L score for about 0.1 points on the Gigaword dataset. Although without a significant boost on the ROUGE-L score, we notice clear differences before and after applying the DUP function, as the examples in Table 1 help illustrating.

| Before | After | Reference |
|--------|-------|-----------|
| bosnian pm's resignation provokes political political political crisis | bosnian pm's resignation provokes political turmoil | prime minister's resignation throws bosnia into crisis with yugoslavia |
| sandelin sandelin sandelin wins spanish open | sandelin wins spanish open | sandelin wins spanish open eds: adds quotes from sandelin and spence |
| credit markets subdued amid stress stress crisis | credit markets subdued amid stress fears | difficult credit markets show strained banking system |
| spanish 'belle rafael rafael azcona dies at 81 | spanish 'belle rafael azcona dies at 81 | spanish 'belle epoque' scriptwriter rafael azcona dies aged 81 |
| nigerian productivity award licence licence | nigerian productivity award licence can be withdrawn | productivity award can be revoked, says nigerian official |
| sports column : the big big big big big big big ap photo <UNK> | sports column : the big league is a big place | in the big 12, basketball does the muscle flexing |

Table 1: Examples of the impact of the DUP function on model output.