[Reviews · NeurIPS 2017]

Reviewer 1



The paper presents a new method for structured output prediction using reinforcement learning. Previous methods used reward augmented maximum likelihoods or policy gradients. The new method uses a soft-max objective. The authors present a new inference method that can be used to efficiently evaluate the integral in the objective. In addition, the authors propose to use additional reward functions which encode prior knowledge (e.g. to avoid word repetitions). The paper seems to be promising from the sequence learning point of view as it can produce sequences of considerably higher quality. However, the authors seem to be ignorant to a large part of the reinforcement learning and policy search literature, please see [1,2] for good reviews of policy search with information-geometric constraints. (1) The soft-max objective is common in the reinforcement learning literature. The given objective is the dual of the following optimisation problem: J = E_{p}[R] + KL(p||p_\theta). If we solve that for p we exactly get the sampling distribution q given in the paper. Plugging the solution back in the objective J yields the dual problem of the objective which is equivalent to equation 7. I think such insights reveal much of the structure of the optimisation problem. Moreover, please note that he objective J is commonly used in policy search, see for example the REPS algorithm, KL control, linearly solvable MDPs, trust region policy optimisation etc... There is an interesting difference though, that commonly, we want to compute p and p_theta is the sampling distribution. In this algorithm, we optimise for p_theta, i.e., we change the sampling distribution. I think this connections to standard policy search algorithms need to be revealed in this paper to provide a better understanding what the algorithm actually does. (2) The RAML algorithm is a special case of expectation-maximisation based policy search while the new algorithm is an instance of information-theoretic policy search, see [1]. Clarifying these relations should help to improve the paper significantly. While in general, I like the approach, I have one technical concern regarding the sampling procedure: The weight w_i for the reward function is either 0 (with probability p_drop) or W. The authors argue that this is equivalent to using a weight that is given by p_drop * W. However, this is incorrect as E_p(drop)[exp(log p(z) + I(drop == 0) * W * reward)] is *not* the same as exp(E_p(drop)[log p(z) + I(drop == 0) * W * reward)] = exp(log p(z) + p_drop * W * reward)] In fact, the distribution the authors are sampling from is rather a mixture model: p_drop * p(z) + (1-p_drop) * exp (W * reward) / Z This is a very different distribution from the one given in the algorithm so I do not think it is sound to use it. I think this issue need to be addressed before publishing the paper (maybe its also a misunderstanding on my side, but the authors need to address my concerns and either proof my wrong or suggest a sound sampling method). Using KL trust regions similar as in the REPS algorithms could for example be a sounder way of finding the weights w_i). In summary, this is in interesting paper, but I have a concern regarding the soundness of the sampling procedure which needs to be addressed. [1] Deisenroth, Neumann, Peters: A survey on policy search for robotics", FNT 2013 [2] Gergely Neu, Anders Jonsson, Vicenç Gómez, A unified view of entropy-regularized Markov decision processes, 2017

Reviewer 2



This paper studies the exposure-bias problem and the wrong-objective problem in sequence-to-sequence learning, and develop a new softmax policy gradient method, which could learn the model with cold-start. The approach combines the RAML and policy gradient method to effectively exploit the model prediction distribution and the reward distribution. An efficient approximate sampling strategy is also developed for estimate the gradient. The method is evaluated on text summarization task and image caption task, both of which show better performance than previous methods. This paper studies an important problem and presents interesting result. There are some minor points that the authors may clarify: The statement “the large discrepancy between the model prediction … and the reward distribution R(z | y)…” (line 113-114) is not clear to me. Please clarify how to see this from Eq (6). Specifically, what does it mean by “the reward distribution R(z | y)”? A probability distribution of R(z | y)? It is not clear how is the cost function (7) motivated. Although from the gradient formula (8), it can be viewed as a combination between the gradients of (2) and (4), a clearer justification and explanation of the meaning of this cost itself is necessary. For example, why softmax function is chosen to combine the model and reward together? Please clarify more.

Reviewer 3



This paper proposes a variant of the RAML method to train RNNs, with the weights containing both the exponentiated reward and the generative probability. This last terms allows to avoid the need for warm starts (ML pretraining). The paper is well written and the examples shown are convincing. A few points to address: 1) The title is confusing, since it refers to "reinforcement learning" and not to sequence prediction. 2) How is the decoding performed? Beam search?